# Complexation of Bromelain, Ficin, and Papain with the Graft Copolymer of Carboxymethyl Cellulose Sodium Salt and *N*-Vinylimidazole Enhances Enzyme Proteolytic Activity

**DOI:** 10.3390/ijms241411246

**Published:** 2023-07-08

**Authors:** Andrey V. Sorokin, Svetlana S. Goncharova, Maria S. Lavlinskaya, Marina G. Holyavka, Dzhigangir A. Faizullin, Yuriy F. Zuev, Maxim S. Kondratyev, Valeriy G. Artyukhov

**Affiliations:** 1Biophysics and Biotechnology Department, Voronezh State University, 1 Universitetskaya Square, 394018 Voronezh, Russia; andrew.v.sorokin@gmail.com (A.V.S.); olsahannikovas@gmail.com (S.S.G.); maria.lavlinskaya@gmail.com (M.S.L.); holyavka@rambler.ru (M.G.H.); ma-ko@bk.ru (M.S.K.); artyukhov@bio.vsu.ru (V.G.A.); 2Bioresource Potential of the Seaside Territory Laboratory, Sevastopol State University, 33 Studencheskaya Street, 299053 Sevastopol, Russia; 3Kazan Institute of Biochemistry and Biophysics, FRC Kazan Scientific Center of RAS, 2/31 Lobachevsky Street, 420111 Kazan, Russia; dfaizullin@mail.ru; 4Alexander Butlerov Chemical Institute, Kazan Federal University, Kremlevskaya Street 18, 420008 Kazan, Russia; 5Institute of Cell Biophysics of the RAS, 3 Institutskaya Street, 142290 Pushchino, Russia

**Keywords:** bromelain, ficin, papain, complexation, proteolytic activity

## Abstract

This study investigates the features of interactions between cysteine proteases (bromelain, ficin, and papain) and a graft copolymer of carboxymethyl cellulose sodium salt with *N*-vinylimidazole. The objective is to understand the influence of this interactions on the proteolytic activity and stability of the enzymes. The enzymes were immobilized through complexation with the carrier. The interaction mechanism was examined using Fourier-transform infrared spectroscopy and flexible molecular docking simulations. The findings reveal that the enzymes interact with the functional groups of the carrier via amino acid residues, resulting in the formation of secondary structure elements and enzyme’s active sites. These interactions induce modulation of active site of the enzymes, leading to an enhancement in their proteolytic activity. Furthermore, the immobilized enzymes demonstrate superior stability compared to their native counterparts. Notably, during a 21-day incubation period, no protein release from the conjugates was observed. These results suggest that the complexation of the enzymes with the graft copolymer has the potential to improve their performance as biocatalysts, with applications in various fields such as biomedicine, pharmaceutics, and biotechnology.

## 1. Introduction

Cysteine proteases are a family of hydrolytic enzymes characterized by the presence of a cysteine residue in their active site. They are widely distributed across various organisms. For instance, the main protease of the SARS-CoV-2 virus, responsible for the post-translational processing of viral proteins, belongs to this class [1]. However, plant enzymes hold a special position among cysteine proteases due to the abundance of raw materials available for their extraction and their relatively lower cost compared to animal-derived enzymes. They are widely utilized across various industries. For instance, ficin (EC 3.4.22.3) and bromelain (EC 3.4.22.32/3.4.22.33), enzymes derived from *Ficus carica* and *Ananas comosus* plants, respectively, have been employed in the development of antibacterial drugs that target bacterial biofilms and wound exudate [2,3,4]. Papain (EC 3.4.22.2), an enzyme isolated from *Carica papaya*, finds successful application in the field of cosmetology [5,6]. Additionally, these enzymes play a crucial role in the tenderization of protein-rich foods [7,8,9].

As mentioned above, the active site of bromelain, ficin, and papain comprises a catalytic dyad consisting of cysteine and histidine [10]. Furthermore, some researchers have observed the participation of asparagine and glutamine in catalysis. Glutamine contributes to the formation of an oxyanion hole, thereby stabilizing the enzyme-substrate intermediate [10]. The significance of asparagine in catalysis varies among the enzymes. As a result, the primary function of the active site is to increase the acidity of the nucleophilic residue responsible for attacking the substrate [11]. Meanwhile, deprotonation of the cysteine thiol group is observed in some cysteine proteases even before the catalytic process [12]. This active site structure is also utilized by certain amidases, such as *N*-glycanase, which hydrolyzes non-peptide C–N bonds [13].

As previously mentioned, cysteine proteases contain a thiol group in cysteine, which acts as a strong nucleophile [14]. Consequently, these enzymes serve as effective biocatalysts but are also susceptible to environmental factors. Such factors can adversely affect the enzymes’ crucial properties, resulting in reduced proteolytic activity and stability, as well as narrower temperature and pH optima [15]. This issue has garnered the attention of researchers in various fields of biology and chemistry for many years. Numerous approaches have been proposed, ranging from expensive proteomic design and production of mutant enzymes [16,17] to inefficient chemical modification of natural proteins [18]. However, the most common approach to enzyme protection is immobilization on different carriers. Immobilized enzymes have been widely used in industry since the 1960s [19,20], but the selection of an immobilization matrix has traditionally been carried out through empirical methods. This approach hampers the efficient utilization of new biocatalysts in industrial production and the accumulation of theoretical knowledge regarding activity alterations in different microenvironments and enzyme interactions with immobilization matrices.

Our previously studies show that natural polymers, such as chitosan [2,14] and its derivatives [21,22,23,24] or nanoparticles based on it [25,26] are promising carriers for cysteine protease immobilization. This process can lead to enhance of proteolytic activity and make it possible to use immobilized enzymes as antibacterial and antibiofilm agents as well as in wound healing [2,27]. Therefore, we are continuing with new natural carriers for the immobilization of cysteine proteases.

Cellulose is the most prevalent natural polymer, known for its renewable nature, biocompatibility, low toxicity, and hydrophilicity. It is characterized by a high content of hydroxyl groups, which facilitate the formation of an intricate network of intra- and intermolecular hydrogen bonds [28]. As a result, cellulose forms fibrils with varying degrees of order and crystallinity. However, due to this extensive hydrogen bonding, cellulose is insoluble in aqueous media and most organic solutions, thus limiting its practical applications. To overcome this challenge, chemical modification of cellulose macromolecules, such as carboxymethylation, is employed. Carboxymethyl cellulose sodium salt (Na-CMC), the resultant salt, exhibits water solubility and forms highly viscous solutions that, when dried, yield elastic porous films [29].

Na-CMC is considered bioinert and finds utility in surgical procedures to prevent the formation of adhesions [30,31]. Furthermore, physical Na-CMC gels possessing a porous structure serve as excellent matrices for immobilizing cysteine proteases [32]. This immobilization shields the proteases from atmospheric oxygen exposure while facilitating targeted substrate transport to enzyme globules. However, the sorption capacity of carboxymethyl cellulose alone may not be sufficient for the production of effective biocatalysts. Therefore, chemical modification, such as grafting high complexing side chains, presents a suitable solution in such cases. Consequently, the combination of carboxymethyl cellulose sodium salt and *N*-vinylimidazole graft copolymer, characterized by low toxicity, was chosen as the enzyme carrier for the present study [30,31,32,33].

With the advancements in bioinformatics and computational biology, the identification of interaction mechanisms between enzymes and immobilization matrices, as well as their effects on enzyme structure, has become more accessible. Molecular docking methods have made it possible to unveil the interaction mechanisms down to the specific amino acid residue and the characteristic type of interaction [34,35,36,37]. However, due to the imperfections of simulation algorithms and the absence of a complete macromolecule model, in silico results should be validated with experimental data. In this regard, Fourier-transform infrared (FTIR) spectroscopy is a promising method, as it allows for the detection of changes in protein secondary structure [14,38,39,40,41], as well as alterations in intra- and intermolecular interactions with the matrix. Additionally, this method is relatively straightforward in terms of equipment and requires minimal consumables [38,39,40,41].

In light of the aforementioned considerations, the objective of this research is to investigate the interaction mechanism between cysteine proteases (bromelain, ficin, and papain) and a graft copolymer of carboxymethyl cellulose sodium salt and *N*-vinylimidazole (Na-CMC-*g*-PVI), while evaluating the impact of this interaction on the proteolytic activity and stability of the enzymes.

## 2. Results and Discussions

As mentioned earlier, FTIR spectroscopy is an accessible and convenient method for detecting qualitative changes in molecular structures. In the case of protein studies, it also enables the assessment of changes in protein secondary structure. To achieve this, it is necessary to investigate the shifts in the position of the amide I band, which reflects the vibrational modes of the carbonyl groups in the peptide bonds of the proteins [38,39,40,41]. The precise location of the amide I (or amide I’ in deuterated samples) band in the FTIR spectrum is influenced by the hydrogen bonds formed by the oxygen atom of the carbonyl group [39]. By tracking changes in the wavenumber of the amide I band, it is possible to estimate the elements comprising the protein’s secondary structure. Therefore, by combining the information about shifts and changes in the shape of absorption bands corresponding to the functional groups of both the carrier and the protein, one can obtain a comprehensive understanding of the peculiarities involved in the formation of the protein–carrier complex.

In order to assess the potential use of graft copolymers of carboxymethyl cellulose sodium salt and *N*-vinylimidazole as carriers for cysteine proteases, FTIR spectra of the carrier and its conjugates with the enzymes were recorded. Figure 1 shows the FTIR spectrum of the Na-CMC-*g*-PVI copolymer, which exhibits characteristic absorption bands: a band at approximately 916 cm^−1^ corresponding to the deformation vibrations of the imidazole cycles, a broad band with two distinct maxima at 1065 and 1085 cm^−1^ resulting from the combined stretching symmetric skeletal vibrations of the pyranose cycles and the C-O-C bond vibrations, several bands around 1326, 1413, and 1593 cm^−1^ attributed to dissociated carboxyl groups (ρCH_2_, ν_s_COO^−^, and ν_as_COO^−^, respectively), and bands near 1467 and 1385 cm^−1^ corresponding to methylene group vibrations [24,30,31,42].

The FTIR spectra of the conjugates between cysteine proteases and the carrier exhibit these characteristic bands, along with additional bands that arise from the peptide structure, such as the amide I’ band (~1622–1643 cm^−1^) [40,41]. Furthermore, a decrease in the intensity of bands related to dissociated carboxyl groups and imidazole cycles is observed, as well as changes in the shape and wavenumbers of bands associated with pyranose cycles and C–OH bonds. These observations indicate the interaction of enzymes with the carrier matrix through the aforementioned functional groups. It should be noted that the FTIR results are consistent with the findings obtained from flexible molecular docking. The affinity, represented by changes in the interaction energy between the carrier and the enzyme, exhibits negative values (Table 1), indicating a spontaneous and thermodynamically favorable interaction between the components of the system.

Evaluation using the modified Lowry method reveals that the enzyme content in the conjugates is approximately equal: 32.8 ± 1.5 mg × g^−1^ for bromelain, 30.0 ± 1.8 mg × g^−1^ for ficin, and 37.5 ± 1.1 mg × g^−1^ for papain. Furthermore, the conjugates demonstrate a high protein immobilization yield (Figure 2A).

The proteolytic activity of the immobilized enzymes is comparable to each other and exceeds that of the native enzymes. The activity of bromelain (in U × mL^−1^ of solution) is 156 ± 2.6 and 97 ± 7; ficin is 157 ± 3.2 and 96 ± 2; papain is 137 ± 5.8 and 95 ± 4 for the immobilized and native enzymes, respectively (Figure 2B). However, the specific activity of the immobilized enzymes (in U × mg^−1^ of protein) shows more significant differences, with values of 95 ± 1.5 for bromelain, 103 ± 2.2 for ficin, and 73 ± 3.1 for papain (Figure 2C). This indicates that the immobilized enzymes undergo hyperactivation, meaning they adopt a conformation that is more catalytically favorable compared to the native enzymes. It is worth noting that the specific activity increased almost 2-fold for bromelain and ficin (196% and 215%, respectively), while for papain, the increase was 154%.

To explain these findings, we can refer to the FTIR and flexible molecular docking data. As mentioned earlier, a comparison of the position of the amide I’ band in the spectra of native and immobilized enzymes allows us to draw conclusions about structural changes in proteins. Figure 3 and Table 1 displays the amide I’ bands of native bromelain, papain, and ficin in a deuterated buffer solution, as well as in their conjugates with Na-CMC-*g*-PVI obtained by subtracting the spectra of the buffer and carrier from the spectrum of the conjugates.

As observed from the presented data, the interaction of bromelain and ficin does not result in significant changes in the shape or position of the amide I’ band. The slight increase in the wavenumber of the amide I’ band of ficin and bromelain upon interaction with Na-CMC-*g*-PVI can be attributed to the formation of hydrogen bonds involving amino acids within the α-helices of the enzyme globules [38,39,40,41,43]. These findings are consistent with the molecular docking data (Table 2, Figure 4), which indicate that bromelain and ficin primarily engage in H-bonds and other weak physical interactions with amino acids located in α-helices or disordered regions of the protein globules. The obtained data (Table 1) demonstrate that the interaction of papain with Na-CMC-*g*-PVI leads to a shift in the wavenumber corresponding to β-sheet signals, indicating an augmentation in their content within the enzyme globules [38,39,40,41].

The observations regarding conformational changes in the studied enzyme globules are further supported by the calculated ratios of α-helices and β-structures for native and conjugated proteins (Table 3 and Figure 3). From this point of view, the secondary structure content for bromelain and ficin is essentially the same in free and bound states, but for papain differs significantly. An increase in the abundance of β-structures is observed, leading to substantial deviations in the protein’s secondary structure.

Moreover, as demonstrated by molecular docking, catalytically important amino acids of all enzymes engage in physical interactions, with H-bond formation observed for bromelain and papain. These weak physical interactions involving key catalytic amino acids likely contribute to the significant increase in proteolytic activity and the attainment of a more favorable enzyme conformation. Modulation of active sites by these interactions with catalytically significant amino acids appear to play a role in achieving these improvements.

In most cases, the primary advantage of immobilized enzymes is their enhanced stability compared to their native counterparts. Therefore, the stability of the conjugates obtained was evaluated by measuring the residual catalytic specific activity (in U *×* mg^−^^1^ of protein) after incubating the samples at 37 °C in 50 mM Tris-HCl buffer at pH 7.5. After 1 day, the immobilized enzymes retained over 80% of their catalytic activity. It is worth noting that the stability of native papain is comparable to that of the immobilized form, unlike bromelain and ficin. After 5 days, and with further incubation, the differences in the loss of catalytic ability between enzymes in solution and those conjugated with the carrier became more pronounced: the immobilized enzymes retained up to 75% of their activity, while the native enzymes retained no more than 40%. The most significant stabilization of catalytic activity was observed after 21 days of incubation: the residual specific activity of ficin did not exceed 7%, whereas its complex with the Na-CMC-*g*-PVI copolymer retained over 50% of the initial activity (Figure 5).

The release of enzymes from the conjugates with the carrier has a significant impact on their potential practical applications. Figure 6 illustrates the protein content in the resulting immobilized formulations during 21 days of incubation at 37 °C in 50 mM Tris-HCl buffer at pH 7.5. As observed from the data presented, the protein content in the immobilized preparations remains nearly constant, indicating a lack of enzyme release. These results are consistent with the molecular docking data, which indicate that the formation of numerous H-bonds and other physical interactions between enzymes and Na-CMC-*g*-PVI contributes to the formation of stable complexes.

It should be noted that the results obtained in this work are rather promising. The phenomenon of hyperactivation in cysteine proteases is quite rare, which calls us for further investigation of these systems using additional instrumental methods in biophysics. The resulting complexes demonstrate sufficient stability in terms of protein content and protease activity, making them promising candidates for testing as antibiofilm agents and in wound care.

## 3. Materials and Methods

### 3.1. Materials

Cysteine proteases, namely bromelain (B4882), papain (P4762), and ficin (F4165), supplied by Sigma-Aldrich, Munich, Germany, were selected as the subjects of this research. Azocasein purchased from Sigma-Aldrich, Munich, Germany, was used as the hydrolysis substrate in the evaluation assays for catalytic activity. Carboxymethyl cellulose sodium salt with a molecular weight of 90,000 and a degree of substitution of 0.7, along with *N*-vinylimidazole, were both acquired from Sigma-Aldrich, Munich, Germany for graft copolymer synthesis. The monomer was purified through vacuum distillation before use and was characterized by the following parameters: bp = 78–79 °C/11 mm Hg; *n*^20^*_D_* = 1.5338.

### 3.2. Synthesis and Characterization of the Enzyme Carrier

The graft copolymer of carboxyethyl cellulose sodium salt with *N*-vinylimidazole (Na-CMC-*g*-PVI) was chosen as the matrix for enzyme immobilization. The Na-CMC-*g*-PVI graft copolymer was synthesized using the following procedure: 0.50 g of carboxymethyl cellulose sodium salt was dissolved in 85 mL of distilled water. Then, 0.05 g of potassium persulfate was dissolved in 5 mL of distilled water, and this solution was added to the polysaccharide. The mixture was degassed through three freeze–thaw cycles, followed by the addition of 0.04 g of sodium metabisulfite under an argon flow. The reactor was placed in a water bath and maintained at 40 °C for 20–30 min. Subsequently, 5.34 mL of *N*-vinylimidazole, which had been pre-degassed, was added to the reaction mixture under an argon flow. The final reaction mixture with a volume of 100 mL was kept at 40 °C for 18 h. The resulting product was isolated via precipitation in acetone, followed by centrifugation and drying in a vacuum oven at 55 °C until a constant weight was achieved. The Na-CMC-*g*-PVI copolymer obtained was further purified from impurities using ethanol in a Soxhlet extractor. Purification was monitored spectrophotometrically. The obtained copolymer and grafted PVI chains were characterized as described in [30]. The grafting efficiency was found to be 42%, and the molecular weight (Mw) of the grafted chains was determined to be 12,569.

### 3.3. FTIR

For analysis, Na-CMC-*g*-PVI copolymer and its conjugates with enzymes were washed with buffer solutions in D_2_O. Solutions and solid wet samples were placed on the surface of the attenuated total reflectance (ATR) working element and maintained at 25 °C. FTIR spectra of the samples were recorded using an IRAffinity1 spectrometer (Shimadzu Scientific Instruments, Kyoto, Japan) equipped with an ATR attachment with a single reflection ZnSe working element and a resolution of 4.0 cm^−1^. Spectra of Na-CMC-*g*-PVI and D_2_O buffer solutions were subtracted from the spectra of the conjugates to isolate the pure amid I’ absorbance of the enzymes. To obtain protein spectra from solutions, the buffer spectra were subtracted. Second-derivative spectra were obtained using a five-point window. The band positions obtained from the second derivative were used as the initial guess for curve fitting of the amid I’ band in the original spectra using the fitting routine Fityk 0.9.8 (http://fityk.sharewarejunction.com/, accessed on 2 June 2023). The following amide I band mean peak positions (cm^−1^) were used for protein pecondary structure motifs: α-helix—1648–1657, β-sheet—1623–1639, β-turn—1660–1680, and random coil—1641–1646 [38,39,40,41].

### 3.4. Molecular Docking

The preparation of the structures of bromelain (PDB ID: 1W0Q, https://www.rcsb.org/structure/1W0Q, accessed on 2 June 2023), ficin (PDB ID: 4YYW, https://www.rcsb.org/structure/4YYW, accessed on 2 June 2023), and papain (PDB ID: 9PAP, https://www.rcsb.org/structure/9PAP, accessed on 2 June 2023) for docking followed the standard scheme for Autodock Vina (https://sourceforge.net/projects/autodock-vina-1-1-2-64-bit/, accessed on 2 June 2023), as described by the software authors on the website. Solvent, buffer, and ligand atoms and coordinates were removed from the input PDB file. The center of the molecule and box parameters were manually set to ensure that the protease molecule was completely within the computational space domain [14].

The structure model of the Na-CMC-*g*-PVI copolymer was created using the molecular constructor HyperChem (https://hyperchem.software.informer.com, accessed on 2 June 2023) and successively optimized in the AMBER force field and then quantum-chemically in PM3 (Parametric Method 3). The ligand in docking calculations had maximum conformational freedom, allowing rotation of functional groups around all single bonds. The arrangement of charges on the Na-CMC-*g*-PVI copolymer molecule and its protonation/deprotonation were performed automatically using the MGLTools 1.5.6 package (https://ccsb.scripps.edu/mgltools/1-5-6, accessed on 2 June 2023).

Sequential docking, also known as cascade or multiple docking, was employed to obtain in silico results. The process involved finding the optimal position of the ligand in the first stage, where its structure was then fixed at the docking site and became an integral part of the target. This binding site, characterized by the highest affinity to the ligand, was blocked. The position of the second ligand was then simulated, and the second Na-CMC-*g*-PVI molecule in the optimal position on the globule also became a part of the target. In the third stage, both bound ligands became part of the receptor. This iterative searching and filling of the optimal binding sites on the target surface were repeated until the positions of all five ligands were simulated.

### 3.5. Enzyme Immobilization

Enzyme immobilization on the Na-CMC-*g*-PVI copolymer was performed as follows: 20 mL of an enzyme solution with a concentration of 2 mg × mL^−1^ in 50 mM glycine buffer at pH 9.0 was added to 1 g of the polymer and incubated for 2 h. After incubation, the gel-like precipitate was purified from unbound enzymes via dialysis using a cellophane bag with a cutoff of 25 kDa against 400 mL of 50 mM Tris-HCl buffer at pH 7.5. Purification was continued until no protein was detected in the washing water, and it was monitored spectrophotometrically using an SF-2000 spectrophotometer (λ = 280 nm, LOMO-Microsystems, Saint-Petersburg, Russia).

### 3.6. Protein Content Assay

The protein content in the enzyme conjugates with the Na-CMC-*g*-PVI copolymer was determined using the Lowry method [44] with the following modification: during the initial stage of the analysis, the bonds between the modified polysaccharide and the enzyme molecules were broken. For this purpose, immobilized enzymes were treated with a 0.7 M solution of K, Na-tartrate prepared with 1 M NaOH at 50 °C for 10 min [45].

### 3.7. Proteolytic Activity Assay

The proteolytic activity of the conjugated enzymes was assessed using the azocasein substrate [46]. The unit of catalytic activity was defined as the amount of enzyme that catalyzes hydrolysis of 1 μM of the substrate in 1 min under the experimental conditions.

### 3.8. Data Statistical Processing

All the experimental studies were carried out with at least eight repetitions. Statistical processing of the results was carried out using the Stadia 8.0 Professional software package (http://protein.bio.msu.ru/~akula/Podr2~1.htm, accessed on 2 June 2023). The statistical significance of differences between the control and experimental values was determined using Student’s *t*-test (at *p* < 0.05), since all indicators were characterized by a normal distribution.

## 4. Conclusions

In conclusion, our investigation into the conjugation mechanism of cysteine proteases (bromelain, ficin, and papain) with the graft copolymer of carboxymethyl cellulose sodium salt and *N*-vinylimidazole has revealed several key findings. First and foremost, it is important to note that the interaction between the studied enzymes and the synthesized carrier results in an enhancement of their catalytic capacity which, however, is different for papain compared to the other two enzymes studied. Among the enzymes studied, papain demonstrates the highest number of interactions formed through catalytically important amino acids. Furthermore, the conjugation of enzymes with Na-CMC-*g*-PVI induces some changes in protein secondary structure. For bromelain and ficin, a structure preservation is observed, while papain undergoes an increase in β-sheet content. These conformational changes, along with weak physical interactions between catalytically significant amino acids and functional groups of the matrix, result in enzyme hyperactivation.

Interactions of enzymes with Na-CMC-*g*-PVI cause an increase in stability, retaining over 50% of their specific proteolytic activity after 21 days of incubation at 37 °C in 50 mM Tris-HCl buffer at pH 7.5. Importantly, the protein content in the enzyme conjugates remains practically unchanged, indicating a lack of enzyme release. The resulting complexes demonstrate sufficient stability in terms of protein content and protease activity, making them promising candidates for testing as antibiofilm agents and in wound care.

## Figures and Tables

**Figure 1 ijms-24-11246-f001:**
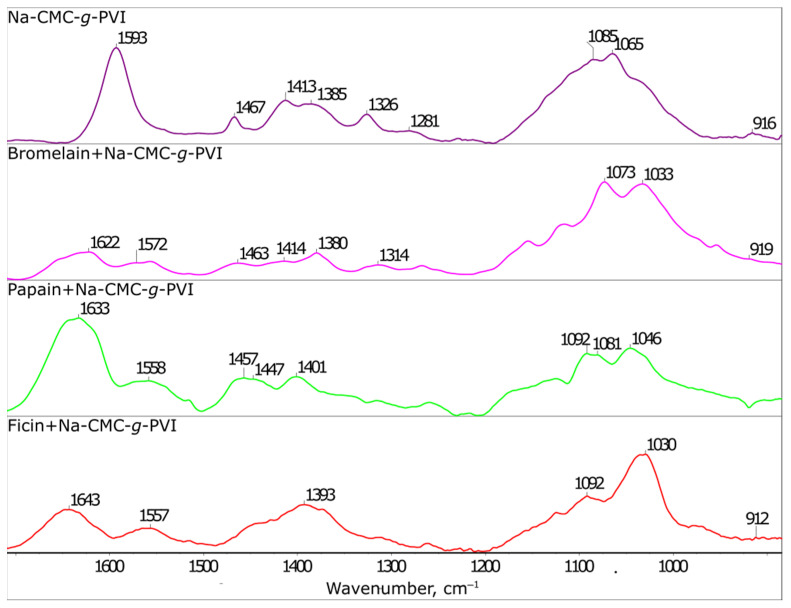
FTIR spectra of the Na-CMC-*g*-PVI copolymer and its conjugates with bromelain, ficin, and papain in borate D_2_O buffer with pH 9.

**Figure 2 ijms-24-11246-f002:**
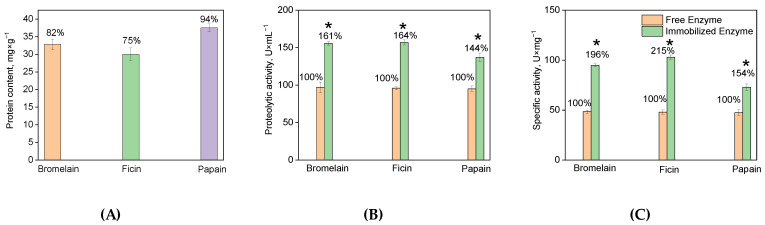
Characteristics of immobilized ficin, papain, and bromelain: (**A**) shows the protein content (mg × g^−1^ of carrier); (**B**) shows the total activity (U × mL^−1^ of solution); (**C**) shows the specific activity (U × mg^−1^ of protein). The efficiency of complexation for each protease is expressed as a percentage of the sorbed enzyme from its amount in solution (**A**), the total activity (**B**), and specific activity (**C**) of the immobilized enzyme compared to the soluble enzyme, indicated above the bars. Note: * statistically significant difference from the soluble enzymes (*p* < 0.05; *n* = 8).

**Figure 3 ijms-24-11246-f003:**
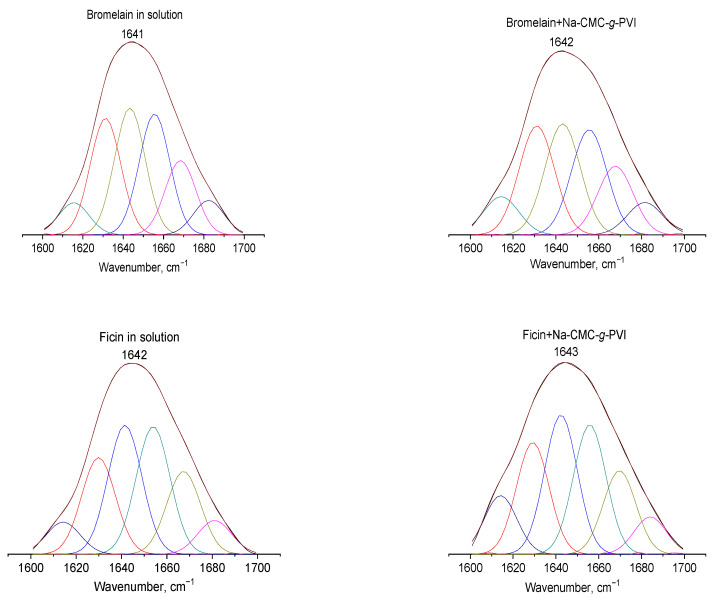
Positions of amide I’ bands of free enzymes or their conjugates in borate D_2_O buffer with pH 9 and curve fitting. Curves for secondary structure elements are in different colors.

**Figure 4 ijms-24-11246-f004:**
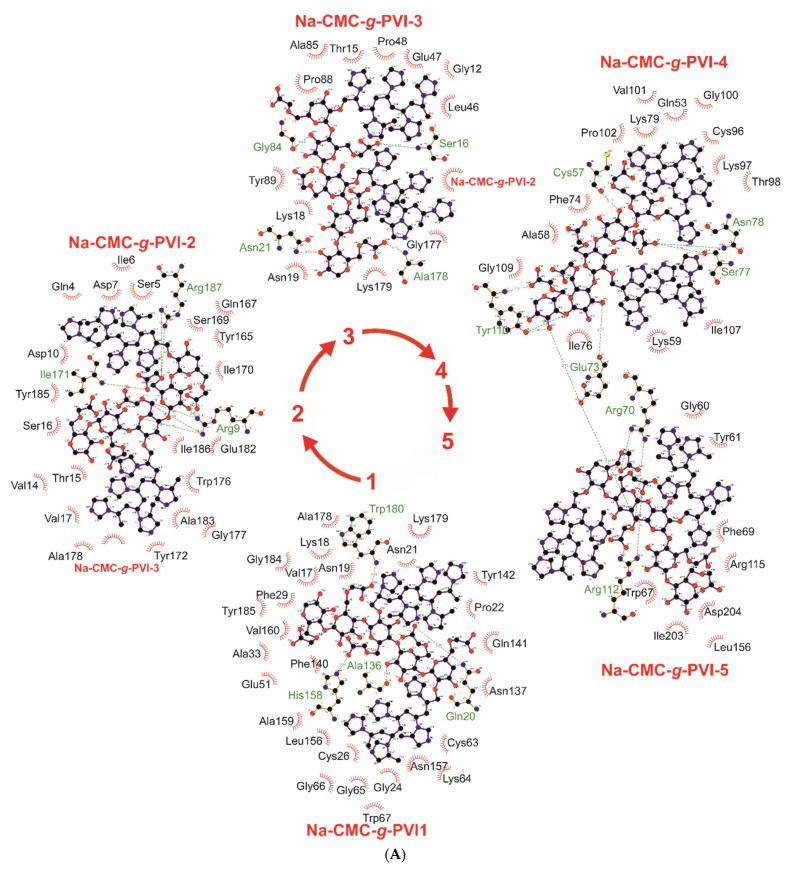
Hydrogen bonds (indicated by dashed lines) and interactions between the molecules of bromelain (**A**), ficin (**B**), and papain (**C**) and the graft copolymer of carboxymethyl cellulose sodium salt and *N*-vinylimidazole simulated by molecular docking.

**Figure 5 ijms-24-11246-f005:**
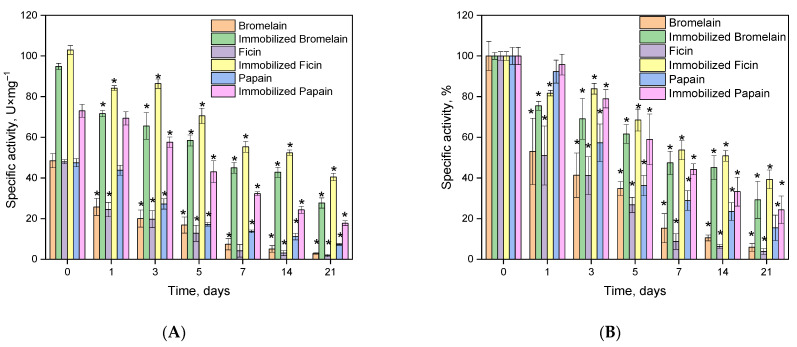
Remaining specific activity of the native or immobilized enzyme in U × mg^−1^ (**A**) or in % (**B**) after incubation at 37 °C in 50 mM Tris-HCl with pH 7.5. Note: * statistically significant difference from the control value without any incubation (*p* < 0.05; *n* = 8).

**Figure 6 ijms-24-11246-f006:**
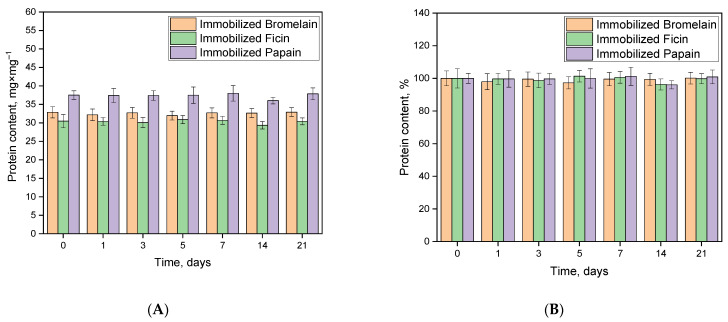
Dependency of the protein content in the immobilized formulations over time in mg × g^−1^ (**A**) or in % (**B**).

**Table 1 ijms-24-11246-t001:** Amide I’ wavenumber, cm^−1^, of free or conjugated enzymes.

Enzyme	Enzyme in Solution	Conjugated Enzyme
Bromelain	1641	1642
Ficin	1642	1643
Papain	1645	1639

**Table 2 ijms-24-11246-t002:** Amino acid residues of the enzymes interacting with the Na-CMC-*g*-PVI copolymer simulated by molecular docking *.

Binding Site Number	Affinity, kcal/mol	Amino Acid Residues Forming	
		H-Bonds, Length, Å	Other Interactions
Bromelain amino acids interacting with Na-CMC-*g*-PVI			
1	−7.0	**Gln20**, 3.07; Ala136, 2.64; **His158** (βR), 3.05; Tpr180, 3.32	Val17, Lys18, Asn19, **Gln20**, Asn21, Pro22, Gly24, **Cys26** (αL1), Phe29 (αL1), Ala33 (αL1), Glu51 (αL2), Cys63, Lys64, Gly65, Gly66, Trp67, Ala136, Asn137, Phe140, Gln141, Tyr142, Leu156, Asn157, **His158** (βR), Ala159 (βR), Val160 (βR), Ala178, Lys179, Tpr180, Gly184, Tyr185
2	−5.5	Arg9, 2.67; 2.79; 3.19 and 3.15; Ile171, 3.08; Arg187, 3.10 and 3.10	Gln4, Ser5, Ile6(βR), Asp7(βR), Arg9, Asp10, Val14, Thr15, Ser16, Val17, Tyr165(βR), Gln167, Ser169, Ile170, Ile171, Tyr172, Gly177, Trp176, Ala178, Glu182, Ala183, Tyr185, Ile186, Arg187
3	−4.9	Ser16, 3.12; Asn21, 3.23; Gly84, 3.00; Ala178, 3.10	Gly12, Thr15, Ser16, Lys18, Asn19, Asn21, Leu46, Glu47, Pro48, Ala85, Pro88, Tyr89, Gly177, Ala178, Lys179
4	−5.4	Cys57 (αL2), 3.10; Glu73 (αL3), 2.77 and 2.79; Ser77 (αL3), 2.78; Asn78, 3.08; Tyr110, 3.06, 3.20 and 3.21	Gln53 (αL2), Cys57 (αL2), Ala58, Lys59, Glu73, Phe74, Ser77, Asn78, Lys79, Cys96, Lys97, Thr98, Gly100, Val101, Pro102, Ile107, Gly109 (βR), Tyr110 (βR), Ile176
5	−5.4	Arg70 (αL3), 3.01 and 2.80; Glu73 (αL3), 2.91; Arg112, 3.31 and 2.94	Gly60, Tyr61, Trp67, Phe69 (αL3), Arg70 (αL3), Glu73 (αL3), Arg112, Arg115, Leu156, Ile203, Asp204
Ficin amino acids interacting with Na-CMC-*g*-PVI			
1	−5.7	Glu145 (αR2), 2.77 and 3.05; Asp161, 3.34 and 2.77	Asn18, Gly20, Arg21, Cys22, Gly23, Tyr60, Cys65, Ser66, Gly67, Gly68, Trp69, Gly140, Glu145 (αR2), Leu146, Lys148, Leu160, Asp161, **His162** (βR), Trp184, Asn187, Trp188
2	−4.8	-	Arg8, Asn14, Val13, Pro15, Ile16, Arg17, Leu45, Ser47, Gln68, Ser87, Pro90, Tyr91, Thr92, Tyr193 (βR)
3	−5.8	-	Asn18, Arg21, Ala93, Lys94, Lys148, Gly185, Thr186, Asn187, Gly189, Arg191
4	−4.7	Asp55, 3.03; Glu97, 2.97	Asn88, Tyr89, Pro90, Gly96, Glu97, Cys98, Asn99, Lys100, Asp101, Leu102
5	−4.5	Cys65, 2.83; Ser66, 2.71 and 2.95; Lys95, 2.92 and 3.15	Arg21, Tyr60, Leu63, Cys65, Ser66, Lys94, Lys95
Papain amino acids interacting with Na-CMC-*g*-PVI			
1	−6.2	Asn64, 3.23; Val157, 3.00; Ser205, 2.99 and 2.85	Asn18, **Gln19**, Gly20, Ser21, Cys23, **Cys25** (αL1), Asn64, Gly65, Gly66, Tyr67 (αL3), Pro68 (αL3), Val133 (βR), Ala136, Ala137, Gln142 (αR2), Leu143 (αR2), Asn155, Lys156, Val157, Asp158 (βR), **His159** (βR), Ala160 (βR), **Trp177**, Gly180, Trp181, Ser205
2	−4.8	**Gln19**, 2.92; Arg83, 2.93; Asn84, 3.79	Val13, Thr14, Pro15, Lys17, Asn18, **Gln19**, Gly20, Ser21, Cys22, Asn46, Gln47, Arg83, Asn84, Pro87, Tyr88, Glu89, Gly90, Thr179, Gly180
3	−5.5	-	Cys63, Asn64, Val91, Gln92
4	−4.5	Arg8, 3.23; Val13, 2.85 and 3.09; Asn184, 3.07 and 3.24	Arg8, Val13, Thr14, Pro15, Thr179, Gly180, Asn184, Tyr186 (βR)
5	−4.9	Arg93, 3.31	Asn84, Glu89, Gln92, Arg93, Tyr94, Arg96

*—catalytically valuable amino acid residues are bold; protein secondary structure elements are in brackets.

**Table 3 ijms-24-11246-t003:** The secondary structure (% of total) of the enzymes, conjugated with Na-CMC-*g*-PVI or in solution.

Structure Elements	Enzyme in Solution	Conjugated Enzyme
Bromelain		
α-helices	24 ± 2	25 ± 2
β-sheets	23 ± 2	26 ± 4
other	53 ± 2	49 ± 4
Ficin		
α-helices	26 ± 2	20 ± 5
β-sheets	22 ± 2	30 ± 6
other	52 ± 2	50 ± 6
Papain		
α-helices	33 ± 2	31 ± 2
β-sheets	19 ± 2	27 ± 11
other	48 ± 2	42 ± 10

## Data Availability

The data presented in this study are available on request from the corresponding author. The data are not publicly available as they are part of the continuous study.

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
