# Peer review of "Complexation of Bromelain, Ficin, and Papain with the Graft Copolymer of Carboxymethyl Cellulose Sodium Salt and N-Vinylimidazole Enhances Enzyme Proteolytic Activity"

_ijms, 2023, doi:10.3390/ijms241411246_

Round 1
Reviewer 1 Report
The authors describe enzyme immobilization on a copolymer of carboxymethyl cellulose and N-vinylimidazole with cysteine proteases. The work will be of interest to those that work in biocatalysis fields. This reviewer recommends publication after some major revisions.
1. Figure 1: This data yields much discussion as to the secondary structural changes upon immobilization. The figure needs to show changes of each enzyme rather than putting them all together. This can be done by separating A and B into separate figures and expanding the data in B to compare the individual enzymes in separate graphs.
2. Table 2 and 3 need to move up to where the FTIR data is being discussed. Also inclusion of the fitting is important here. Show the gaussians and how they represent each type of secondary structure.
3. The docking doesn't tell us much and Figure 3 and Table 1 are confusing. It is unclear what is being conveyed in Figure 3.
My overall recommendation is to take out the docking since it does not add substance to your findings. Rather concentrate on the FTIR fits and the changes in secondary structure, activity of the enzymes over time, and their stabilities. This will significantly improve the manuscript.
Author Response
The authors describe enzyme immobilization on a copolymer of carboxymethyl cellulose and N-vinylimidazole with cysteine proteases. The work will be of interest to those that work in biocatalysis fields. This reviewer recommends publication after some major revisions.
Comment 1. Figure 1: This data yields much discussion as to the secondary structural changes upon immobilization. The figure needs to show changes of each enzyme rather than putting them all together. This can be done by separating A and B into separate figures and expanding the data in B to compare the individual enzymes in separate graphs.
Response: Fig. 1 was modified according your suggestion
Comment 2. Table 2 and 3 need to move up to where the FTIR data is being discussed. Also, inclusion of the fitting is important here. Show the gaussians and how they represent each type of secondary structure.
Response: The required figure was added (see Fig. 3).
Comment 3. The docking doesn't tell us much and Figure 3 and Table 1 are confusing. It is unclear what is being conveyed in Figure 3.
My overall recommendation is to take out the docking since it does not add substance to your findings. Rather concentrate on the FTIR fits and the changes in secondary structure, activity of the enzymes over time, and their stabilities. This will significantly improve the manuscript.
Response: In modern research, especially experimental, the most important goal is to understand the mechanisms that underlie the formation of intermolecular complexes. In our work, it is very important to understand which atomic groups form hydrogen bonds and are responsible for hydrophobic interactions, because the formation of complexes enhances the proteolytic activity of the studied enzymes. Perhaps the interactions cover the zone of the active center? Does the studied ligand act as a substrate mimetic? Or as a cofactor? Answering these questions requires exact atomic coordinates of binding sites, with angles and distances between atoms. Of course, X-ray diffraction or NMR or spectral methods would be ideal for analyzing the resulting structures, but modern computer chemistry methods also make it possible to obtain a picture of interactions. The flexible docking method was created precisely to obtain promising complexes between receptors (for example, proteins) and ligands. Complete freedom of rotation of single bonds and a long enumeration of protein + ligand conformational variants make it possible to analyze first automatically, then rank the best results, and then analyze many variants of cysteine protease complexes and graft copolymer already by a human. The literature describes the strengths and weaknesses of docking methods in sufficient detail, and it is widely used in the virtual screening of drugs and new materials for biomedicine.
Thanks for your work!
Reviewer 2 Report
In this investigation by Y. F. Zuev et al., the authors report the immobilization of Bromelain, Ficin, and Papain with a copolymer of carboxymethyl cellulose sodium salt and N-vinylimidazole. The interactions were examined using FT-IR spectroscopy and flexible molecular docking simulations. Overall the immobilized enzymes demonstrated superior stability compared to their native counterparts, opening opportunities for applications in various fields such as pharmaceutics and biotechnology.
The investigation is not very original (especially as regard to the previous work performed by the corresponding author), but after careful analysis and consideration, I recommend publication in Int. J. Mol. Sci. after the following minor revisions have been performed.
‒ Hyphens of references (i.e., [XX-YY] in the text, and reference section year, vol, XXXX-YYYY) must be inserted via the insert symbol command.
‒ L124, N of N-vinylimidazole must be italicized.
‒ Table 1, Table 2, and Figure 3: In the title, please add more informations. Actually it was calculated via ???
Author Response
In this investigation by Y. F. Zuev et al., the authors report the immobilization of Bromelain, Ficin, and Papain with a copolymer of carboxymethyl cellulose sodium salt and N-vinylimidazole. The interactions were examined using FT-IR spectroscopy and flexible molecular docking simulations. Overall, the immobilized enzymes demonstrated superior stability compared to their native counterparts, opening opportunities for applications in various fields such as pharmaceutics and biotechnology.
The investigation is not very original (especially as regard to the previous work performed by the corresponding author), but after careful analysis and consideration, I recommend publication in Int. J. Mol. Sci. after the following minor revisions have been performed.
Comments of reviewer:
‒ Hyphens of references (i.e., [XX-YY] in the text, and reference section year, vol, XXXX-YYYY) must be inserted via the insert symbol command.
‒ L124, N of N-vinylimidazole must be italicized.
‒ Table 1, Table 2, and Figure 3: In the title, please add more information. Actually it was calculated via ???
Response: the required corrections were performed
Thanks for your work!
Reviewer 3 Report
The paper entitled „Complexation of Bromelain, Ficin, and Papain with the Graft Copolymer of Carboxymethyl Cellulose Sodium Salt and N-Vinylimidazole Enhances Enzyme Proteolytic Activity” focuses on the interaction mechanism between cysteine proteases (bromelain, ficin, and papain) and a graft copolymer of carboxymethyl cellulose sodium salt and N-vinylimidazole, while evaluating the impact of this interaction on the proteolytic activity and stability of the enzymes.
The paper is well-written and interesting. Although the results are understandably submitted, the Introduction section, “Results and Discussion” and Conclusion part should be improved.
I would like to recommend the publication of the paper publication after some changes concerning the following issues:
1. In the introduction section, more information about similar up-to-date approaches to immobilization of cysteine proteases on various substrates together with their pros and cons should be included.
2. The resolution of Figure 1 should be improved. It can be also made larger.
3. Please, improve the caption of Table 2;
4. The results illustrated in Figure 5 can be further discussed;
5. The statistical significance of differences is not mentioned in figures 2, 4 and 5.
6. The limitations of the study could be emphasized at the end of the “Results and Disscution” part;
7. The conclusion section could be improved. It sounds more like a summary of the results than a conclusion.
None
Author Response
The paper entitled „Complexation of Bromelain, Ficin, and Papain with the Graft Copolymer of Carboxymethyl Cellulose Sodium Salt and N-Vinylimidazole Enhances Enzyme Proteolytic Activity” focuses on the interaction mechanism between cysteine proteases (bromelain, ficin, and papain) and a graft copolymer of carboxymethyl cellulose sodium salt and N-vinylimidazole, while evaluating the impact of this interaction on the proteolytic activity and stability of the enzymes.
The paper is well-written and interesting. Although the results are understandably submitted, the Introduction section, “Results and Discussion” and Conclusion part should be improved.
I would like to recommend the publication of the paper publication after some changes concerning the following issues:
Comment 1. In the introduction section, more information about similar up-to-date approaches to immobilization of cysteine proteases on various substrates together with their pros and cons should be included.
Response: Relevant information was added to the Introduction section
Comment 2. The resolution of Figure 1 should be improved. It can be also made larger.
Response: Required corrections were performed.
Comment 3. Please, improve the caption of Table 2;
Response: Required corrections were performed.
Comment 4. The results illustrated in Figure 5 can be further discussed;
Response: We tried to enhance of Fig. 5 description.
Comment 5. The statistical significance of differences is not mentioned in figures 2, 4 and 5.
Response: Required corrections were performed: statistically-differenced values were marked by asterisk symbols.
Comment 6. The limitations of the study could be emphasized at the end of the “Results and Disscution” part;
Response: The required information was added to the manuscript
Comment 7. The conclusion section could be improved. It sounds more like a summary of the results than a conclusion.
Response: We try to rewrite Conclusion to better reflect impact of our work
Thanks for your work!
Round 2
Reviewer 3 Report
The authors have carefully addressed the reviewer's recommendations.
None